# The association of red and processed meat with gestational diabetes mellitus: Results from 2 Canadian birth cohort studies

Rosain N. Stennett[1], Hertzel C. Gerstein[2,3], Shrikant I. Bangdiwala[1,2], Talha Rafiq[4], Koon K. Teo[2,3], Katherine M. Morrison[5,6], Stephanie A. Atkinson[5,6], Sonia S. Anand[1,2,3], Russell J. de Souza[1,2]*

1 Department of Health Research Methods, Evidence & Impact, McMaster University, Hamilton, Ontario, Canada, 2 Population Health Research Institute, Hamilton Health Sciences, Hamilton, ON, Canada, 3 Department of Medicine, McMaster University, Hamilton, ON, Canada, 4 Faculty of Health Sciences, Medical Sciences Graduate Program, McMaster University, Hamilton, ON, Canada, 5 Department of Pediatrics, McMaster University, Hamilton, ON, Canada, 6 McMaster Children's Hospital, Hamilton Health Sciences, Hamilton, ON, Canada

* desouzrj@mcmaster.ca

**Data Availability Statement:** All relevant data are within the paper and its Supporting Information files. Raw data cannot be shared publicly because

## Abstract

### Objective

Red and processed meat is considered risk factors of gestational diabetes mellitus (GDM), but the evidence is inconclusive. We aimed to examine the association between red and processed meat intake and odds of GDM among South Asian and White European women living in Canada.

### Methods

This is a cross-sectional analysis of pregnant women from two birth cohorts: SouTh Asian biRth cohorT (START; n = 976) and Family Atherosclerosis Monitoring In earLY life (FAMILY; n = 581). Dietary intake was assessed using a validated 169-item semi-quantitative food-frequency questionnaire (FFQ). Multivariate logistic regression models were used to examine the associations between gestational diabetes and: 1) total red and processed meat; 2) unprocessed red meat; 3) processed meat and GDM after adjustment for potential confounders.

### Results

There were 241 GDM cases in START and 91 in FAMILY. The median total red and processed meat intake were 1.5 g/d (START) and 52.8 g/d (FAMILY). In START, the multivariable-adjusted odds ratio (OR) showed neither lower nor higher intakes of unprocessed red meat (p-trend = 0.68), processed meat (p-trend = 0.90), or total red and processed meat (p-trend = 0.44), were associated with increased odds of GDM, when compared with medium intake. Similar results were observed in FAMILY except for processed meat intake [OR = 0.94 (95% CI 0.47–1.91), for medium versus low and OR = 1.51 (95% CI 0.77–2.29) for

of lack of participant consent, but are available upon request to Dipika Desai, Program Manager, Population Health Research Institute (PHRI) at: Email: dipika.desai@phri.ca. The Research Ethics Committee that has restricted this data is: Hamilton Integrated Research Ethics Board (HiREB).

**Funding:** START - Canadian Institutes of Health Research (Grant INC-109205; https://cihr-irsc.gc. ca/e/193.html), and the Heart and Stroke Foundation of Canada: (Grant NA 7283; https:// www.heartandstroke.ca/): S S Anand, K Morrison and K K Teo START has not received funding/ assistance from a commercial organization. The funders had no role in study design, data collection and analysis, decision to publish, or preparation of the manuscript.

**Competing interests:** Hertzel C. Gerstein holds the McMaster-Sanofi Population Health Institute Chair in Diabetes Research and Care. He reports research grants from Eli Lilly, AstraZeneca, Novo Nordisk, and Sanofi; honoraria for speaking from AstraZeneca, Boehringer Ingelheim, Eli Lilly, Novo Nordisk, DKSH, Zuellig, Roche, Sanofi, Jiangsu Hanson, and Carbon Brand; and consulting fees from Abbott, Eli Lilly, Novo Nordisk, Sanofi, Kowa, Pfizer, Hanmi and Viatris. Russell J. de Souza has served as an external resource person to the World Health Organization's Nutrition Guidelines Advisory Group on trans fats, saturated fats, and polyunsaturated fats. The WHO paid for his travel and accommodation to attend meetings from 2012-2017 to present and discuss this work. He has also done contract research for the Canadian Institutes of Health Research's Institute of Nutrition, Metabolism, and Diabetes, Health Canada, and the World Health Organization for which he received remuneration. He has received speaker's fees from the University of Toronto, and McMaster Children's Hospital. He has held grants from the Canadian Institutes of Health Research, Canadian Foundation for Dietetic Research, Population Health Research Institute, and Hamilton Health Sciences Corporation as a principal investigator, and is a co-investigator on several funded team grants from the Canadian Institutes of Health Research. He serves as a member of the Nutrition Science Advisory Committee to Health Canada (Government of Canada), a co-opted member of the Scientific Advisory Committee on Nutrition (SACN) Subgroup on the Framework for the Evaluation of Evidence (Public Health England), and as an independent director of the Helderleigh Foundation (Canada). These affiliations do not alter our adherence to PLOS ONE policies on sharing data and materials. START has not received

medium versus high; p-trend = 0.18] after adjusting for additional dietary factors such as the diet quality score, total fiber, saturated fat and glycemic load.

## Conclusion

Medium compared with low or high red and processed meat intake is not associated with GDM in White Europeans and South Asians living in Canada.

## Introduction

Gestational Diabetes Mellitus (GDM) is characterized by elevated blood glucose levels that presents for the first time in pregnancy, typically during the second or third trimester [1,2]. The International Diabetes Federation [IDF] [3] estimates that 12.8% of pregnant women globally are affected by GDM. However, due to regional variation in diagnostic criteria and thresholds, this number has been reported to range from 1.8%– 25.1% [2]. GDM has become the most common pregnancy complication and in Canada, 54.5 per 1,000 deliveries are to women with GDM [4].

GDM is associated with increased risk of obstetrical complications, including pre-eclampsia, pregnancy-induced hypertension, and need for caesarean section [5]. A woman diagnosed with GDM is seven times likely to develop type 2 diabetes mellitus (T2DM) over the 10–15 years following delivery [6]. In utero, GDM is associated with altered fetal growth and development [5,7,8]. Children born to women with GDM are at greater risk of obesity and other abnormal metabolic conditions later in life [6,9].

Ethnicity and diet are risk factors for GDM [10–17]. Women of African, Arab, Asian, Hispanic, Indigenous, Native American, and South Asian ethnicity are at increased risk of developing GDM. Farrar et al. [18], in the UK-based "Born-in-Bradford" birth cohort study among Europeans and South Asian women, found that the prevalence of GDM among South Asians (range 4.1–17.4) was approximately twice that of White British counterparts (range: 1.2–8.7%), irrespective of the diagnostic cut-off threshold used. Anand et al. [19] reported a 36.3% prevalence of GDM among 1006 expectant South Asian mothers in Canada using the ethnic-specific Born-in-Bradford (BiB) criteria. However, there is uncertainty about the exact role/ mechanism of some risk factors such as ethnicity in the development of GDM.

Diet has long been suspected to play an important role in GDM due to its complex effect on established modifiable risk factors. However, the evidence on the association between dietary animal protein sources and GDM is inconsistent. In Southwestern Chinese women, Liang et al. [20] observed that a higher intake compared to low intake of animal protein in mid-pregnancy was associated with increased odds of GDM (RR 1.67 [95% CI: 1.19–2.93], p-trend = 0.03). In a prospective cohort study of pregnant women (> 90% White Caucasian), consumption of animal protein prior to becoming pregnant—in particular total red meat, unprocessed red meat and processed red meat–increased the risk of developing GDM after adjusting for other dietary intake, behavioural factors and body mass index (RR 2.05 [95% CI 1.55–2.73], p = <0.01; 1.60 [95% CI 1.21–2.12], p = <0.01, and 1.36 [95% CI 1.03–1.80], p = 0.06), respectively) [21]. Each additional serving of red meat or processed meat consumed during the year prior to pregnancy increased risk of GDM during the subsequent pregnancy by 60% [22]. These findings are consistent with other studies [23,24]. In contrast, a prospective study of 1,733 pregnant women in the U.S., found that neither red meat nor processed meat intake increased risk of GDM (OR 1.01 [95% CI: 0.91–1.12] and OR 0.95 [95% CI: 0.85–1.06], respectively) [25]. This is of keen interest as national representative cross-sectional survey

funding/assistance from a commercial
organization.

among select North American countries (US, Canada, and Mexico) showed that the median (IQR) intake of unprocessed red meat was higher in Canada (79.0 (36.6–131.6) g), than in the US (72.3 (38.3–124.5) g, p = 0.04) [26]. Furthermore Canadians (79.0 (36.2–140.1) g) consume higher median intakes of total meat when compared to Mexico (62.5 (31.3–117.4) g, p = 0.04), but not different from median intake in the US (79.4 (40.8–134.7) g).

We explored the association of maternal red and processed meat intake with GDM in two Canadian cohorts of single gestation pregnant women (South Asian and White European) age 18–40 years old.

## Methods

We used cross-sectional data from two prospective cohort studies: the SouTh Asian biRth cohorT (START) and the Family Atherosclerosis Monitoring in Early Life (FAMILY). The START study is a prospective birth cohort study among South Asian women (18–40 years of age) from Brampton and Mississauga, Ontario that, commenced in their second trimester of pregnancy. The study seeks to understand the role of fetal programming in the development of adiposity in South Asians in general—a high risk group for T2DM and CVD. In total, 1,012 participants were recruited between 2011 and 2015 and detailed methodology has been previously published [27]. Information concerning medical and pregnancy history, health status, health behaviours, and socioeconomic status was obtained by questionnaires. Anthropometric measurements were collected from mothers at 24–28 weeks of gestation.

The FAMILY study is a prospective birth cohort primarily composed of White Europeans and is designed to understand the early life determinants of cardiovascular disease (CVD) [28]. Between 2004–2009, 859 families (901 babies and 859 mothers) were enrolled. All consenting expectant mothers (between 24–36 weeks of gestation), who planned to have their babies delivered in the Hamilton area, were approached. Mother's demographics, socioeconomic status, health and nutrition, medical and pregnancy history and CVD risk factors were assessed by questionnaires. Following the birth visit, the families were contacted at 6 months.

We opted to use data from these two cohort studies because they represent two different ethnic groups (South Asian and White European), with different background risks for GDM (high and low), and very different dietary patterns. The START cohort included South Asian Canadians, the largest non-white ethnic group in Canada while the FAMILY cohort included White Europeans, the largest ethnic group in Canada, although the FAMILY cohort was multi-ethnic [White European, East or South East Asian, Aboriginal, South Asian, and African or other origins] [28], the majority of participants (>80%) are White European, as a result we opted to exclude other ethnic groups (which includes approximately 2% South Asians) for our analysis. Additionally, both cohort studies assessed diet using a comparable instrument (SHARE FFQs—validated semi-quantitative food-frequency questionnaires), which improved the comparability of dietary data. To account for the intractable association/ confounding with red and processed meat intake and ethnicity (i.e., cohort) we created cohort-specific groups. Finally, these two cohorts encompassed a wide range of meat intake, which we believed would help model the associations more finely.

To be eligible for inclusion in this analysis, a woman had to be pregnant with a single gestation, between 18 and 40 years old, with complete maternal baseline demographic, nutrition, and anthropometric data. Expectant mothers with history of diabetes (T2DM, type 1 DM, and GDM), or who left >10% of items on the FFQ blank, were missing data for both exposures (unprocessed and processed red meat) or reported daily energy intake <500 or >4500 kcals, were excluded. The START cohort consisted of South Asian women only, while the FAMILY cohort consisted of multiple ethnic groups. For the present analysis, we included only White

European participants from FAMILY. After exclusion criteria were applied, the final data set included 1557 participants (START, n = 976 and FAMILY, n = 581) (S1 Fig).

Enrolled participants provided written informed consent for participation, and each study obtained ethics approval from the McMaster Hamilton Integrated Research Ethics Board [START (HiREB #10–640), FAMILY (HiREB #02–060)].

## Assessment of exposure and other dietary variables

The Study of Health and Risk in Ethnic Groups Food Frequency Questionnaires (SHARE-FFQs) was administered at the initial visit in both cohort studies (second to third trimester). The validated semi-quantitative 163-item SHARE-FFQ was developed for South Asians—START (163-item) and White Europeans—FAMILY (157-item) living in Canada. [29]. Briefly, for FFQs, participants are asked to indicate their frequency of consumption of each item over the past 12-months (per day, per week, per month, per year, or never), with three reference serving sizes provided. A picture or common measure was shown for a medium, or typical serving, and a small serving was half the size of a medium, and a large was 1.5 to 2-times the size of a medium [29]. Reproducibility and validity of the SHARE-FFQ can be found in other publications [29–31].

Nutrient intake obtained from the SHARE-FFQs was calculated using the Food Processor nutrient analysis software (version 6.11, 1996, ESHA, Salem, OR), derived from the 1991 Canadian Nutrient File (CFN) and the US Department of Agriculture nutrient food composition databases. The updated 2011 CFN was used to ascertain meat and poultry nutrient values. Recipes were generated for red and processed meat composite food items that were not available in the database, and a nutrient value/ profile was developed. Because serving size varies across food items (e.g., a serving of a hotdog or lunch meat is smaller than a serving of steak or meatloaf) and some items (e.g., ground beef as hamburger, meatloaf, in casseroles) are not entirely meat, the estimated average grams of meat per serving for each food item was calculated and standardized by a registered dietitian familiar with South Asian and European food items. Grams of meat for each sub-item were totalled to get an estimate for each composite item (e.g., baked ham is 100% meat; ground beef as hamburger, meatloaf, in casseroles is 84% meat). Meat groups were defined as follows: "unprocessed red meat" referred to beef, pork, lamb, ham, and veal, and "processed meat" included sausage, salami, bacon, pepperoni, cured, and lunch meat (S1 Table).

Dietary intake for food groups/ items are presented in servings per day and grams per day while nutrient composition is reported in grams per day. Energy adjusted values for nutrients were derived by using the Nutrient Residual Model approach [32]. To describe overall diet quality, we used a previously developed diet quality score (DQS), which uses "the sum of daily servings of "healthy" foods (fermented dairy, fish and seafood, leafy green vegetables, cruciferous vegetables, legumes, fruits, nuts, and whole grains) less the sum of daily servings of "unhealthy foods" (processed meats, refined grains, French fries, snacks, sweets, and sweet drinks)"[33]. Food items listed were selected based on their association with chronic disease risk. High values indicate a healthy dietary pattern, while a negative/low value indicates the increased consumption of unhealthy foods.

## Assessment of covariates

Data collected on maternal characteristics at baseline (24–37 weeks gestation for both cohorts) included: age, pre-pregnancy BMI (pre-pregnancy weight was self-reported and height and weight were measured at study visits to determine BMI), parity, smoking history (categorized as never smoked, quit before this pregnancy, quit during this pregnancy or currently

smoking), physical activity (hours of active sport per week during pregnancy as reported by the mother at her initial visit), family history of DM, marital status (categorized as married/ common-law, never married or divorced/ separated), education (completion of high school or not), and annual household income (<$30,000, $30,000–49 999 or ≥$50,000). Social disadvantage (recoded for this analysis into low, medium and high) was described with a previously validated index, which includes employment, marital status and income [34]. Gestational weight gain was derived from follow-up records at birth, by subtracting the pre-pregnancy weight from the end of pregnancy weight prior to delivery. All missing covariate data were treated as missing at random.

## Assessment of GDM

For our primary analysis, GDM was diagnosed between 24–28 weeks gestation and defined as any one of the following: 1) the mother's self-report of a diagnosis of GDM; 2) the use of insulin during pregnancy (medical record after birth); 3) a birth-chart note of GDM diagnosis; or 4) a positive oral glucose tolerance test (OGTT), using the International Association of the Diabetes and Pregnancy Study Groups [IADPSG] threshold, which was administered to women in both cohorts. The secondary analyses looked at other diagnostic threshold separately: Canadian Diabetes Association [CDA] (75-g OGTT with fasting glucose ≥5.3 mmol/L, 1 hour ≥10.6 mmol/L, 2 hours ≥9.0 mmol/L) and IADPSG threshold (75-g OGTT with fasting glucose ≥5.1 mmol/L, 1 hour ≥10.0 mmol/L, 2 hours ≥8.5 mmol/L) for both cohorts; and the South Asian specific Born-in-Bradford [BiB] threshold (>5.2 mmol/L fasting glucose level, or >7.2 mmol/L or 2-hour post-load) for START. There were no missing data for the primary GDM definition, however, for definitions used in sensitivity analysis, missing data were treated as missing at random.

## Statistical analysis

Dietary intake of red and processed meat, and incidence of GDM varied between cohorts. START participants had >8-fold lesser mean/ median red and processed meat intake than FAMILY participants; and START had 1.6-fold higher incidence of GDM than FAMILY. We therefore decided to examine the data based on cohort-specific intake of red and processed meat.

Participants in each cohort were placed into three equally-size groups according to cohort-specific tertile cut-offs (refers to any of two points of the ordered distribution of consumers which divides the group into three equal parts) for consumption of unprocessed red meat and total red and processed meat (low, medium, and high). Processed meat was handled differently. In START, most participants (56.9%, n = 565/976) reported consuming 0 g/d of processed meat. Therefore, we created a non-consumer group and placed the remaining participants into groups based on tertile cut-off points (low, medium, and high consumers). By creating cohort-specific tertile cut-offs and acknowledging the large number of processed meat non-consumers in the START cohort we minimize the impact of outliers and ensure equal group size at each exposure level, and those optimize our power and stability of estimates within the groups.

Descriptive statistics were used to present participants' characteristics and dietary intake. For trend analysis between exposure and other covariates (demographic characteristics and other dietary intakes) across sub-groups, relevant tests were used for both continuous and categorical variables.

The primary association measure was the odds ratio (OR) of GDM, comparing the highest with the lowest exposure to 1) unprocessed red meat, 2) processed meat, and 3) total red and

processed meat, using multivariable logistic regression, with the low consumption group serving as the reference group. Covariates entered in the models were chosen based on our reading of previous literature and researcher interest and included both biological and lifestyle risk factors for GDM. To build our model, we first used a series of univariate correlation analyses testing variables known or suspected to be associated with either red meat consumption or GDM (e.g. age, pre-pregnancy BMI, family history of DM, etc.). Any variables associated with either exposure our outcome with a univariate correlation p<0.20 were considered for inclusion. Then, we presented models sequentially adjusted for more variables, entered as sets representing an underlying concept (e.g., non-modifiable risk factors; body weight; lifestyle factors; and dietary variables). Five models were used: 1) unadjusted or crude; 2) adjusted for age and parity; 3) additionally adjusted for pre-pregnancy BMI and pregnancy weight gain; 4) additionally adjusted for smoking (in FAMILY only as <1% of participants in START reported smoking), family history of DM, level of education, total energy; 5) additionally adjusted for diet index, total fiber, saturated fat, and glycemic load. Models were assessed for collinearity of independent variables using the variation inflation factor.

All statistical analyses were performed with SAS software (version 9.1; SAS Institute), with values achieving $P < 0.05$ considered statistically significant based on two-tailed test.

## Results

Tables 1 and 2 show the participants' demographic characteristics and dietary factors by intake of total red meat, separately for START and FAMILY (see S2 Table for unprocessed red meat and processed meat). The median intake of unprocessed red meat was 1.34 g/d (0.2–11.0 g/d), of processed meat 0.0 g/d (0.0–0.2 g/d),and of total red and processed meat 1.5 g/d (0.2–11.3 g/d) in START (n = 976). The median intake of unprocessed red meat was 44.1 g/d (27.5–65.9 g/d), of processed meat was 11.5 g/d (6.2–19.7 g/d), and of total red and processed meat was 58.2 g/d (37.1–85.7 g/d) in FAMILY (n = 581). Quarter (24.7% (n = 241)) of START participants and 15.7% (n = 91) of FAMILY participants were diagnosed with GDM (S2 Table). FAMILY participants when compared to START participants were younger and likely to smoke (currently or previously). Within the FAMILY cohort, high total red and processed meat consumers were less likely to be married or in a common-law relationship, and had lower levels of education, employment, and physical activity. START participants who had high total red and processed meat intake were more likely to be unemployed or retired, and have an annual household income >$50,000 compared to their low total red and processed meat consumer counterparts. In both cohorts, high total red and processed meat consumers had a higher pre-pregnancy BMI, and a higher SDI. Higher total red and processed meat consumers were more likely to have a family history of T2D in START.

Red and processed meat consumption was associated with other nutrients and food groups that may influence cardiometabolic health. For example, high total red and processed meat consumers in START consumed higher amounts of total protein, total fat, cholesterol, and less total fibre (Table 2). High total red and processed meat consumers in FAMILY consumed higher total energy, total protein, total fat, saturated fat, cholesterol, and less total carbohydrates, total fibre, and sugar compared to low consumers. In START, higher consumers of total red and processed meat ate more poultry, fish, seafood, and eggs; and less dairy and legumes. In FAMILY, higher consumers of total red meat ate more poultry, fish, seafood, dairy, eggs, and legumes; but lower amounts of nuts.

In START, neither lower nor higher intakes of unprocessed red meat, processed meat, or total red and processed meat were associated with increased odds of GDM, when compared with medium intakes (Table 3), with no evidence of a trend across categories. In FAMILY,

**Table 1. Participant characteristics according to category of dietary intake of total red meat among 581 expectant mothers in the FAMILY cohort and 976 expectant mothers in the START cohort.**

| Variable | FAMILY (n = 581) | | | | | | | START (n = 976) | | | | | | |
|---|---|---|---|---|---|---|---|---|---|---|---|---|---|---|
| | Low | | Medium | | High | | p-trend[†] | Low | | Medium | | High | | p-trend[†] |
| | n | % | n | % | n | % | | n | % | n | % | n | % | |
| **Participants** | 193 | 33.2 | 194 | 33.4 | 194 | 33.4 | - | 325 | 33.3 | 326 | 33.4 | 325 | 33.3 | - |
| **Median Red+Processed meat intake (g/d)** | 30.3 | | 58.2 | | 96.7 | | - | 0.06 | | 1.5 | | 20.5 | | - |
| **Maternal Age (M/SD)** | 31.7 | (4.4) | 31.6 | (4.4) | 31 | (4.7) | 0.10 | 30 | (3.8) | 30.5 | (3.8) | 30. 0 | (4.0) | 0.99 |
| **Pre-pregnancy BMI (kg/m^2) (M/SD)** | 25.5 | (5.5) | 26.1 | (5.8) | 27.4 | (7.2) | <0.001 | 23.4 | (4.1) | 23.6 | (5.6) | 24.3 | (4.8) | 0.01 |
| **Gestational weight gain (kg) (M/SD)** | 14.8 | (5.4) | 15.0 | (4.7) | 14.9 | (5.4) | 0.76 | 14.8 | (9.6) | 14.3 | (7.7) | 13.8 | (5.5) | 0.11 |
| **Maternal Family Hx of DM** | 23.8 | 43 | 17.8 | 32 | 26.6 | 46 | 0.54 | 35.83 | 115 | 43.75 | 140 | 49.84 | 160 | <0.001 |
| **Parity (median/IQR)** | 0 | (1–4) | 1 | (1–4) | 1 | (1–4) | 0.02 | 1 | (1–4) | 1 | (1–3) | 1 | (2–4) | <0.001 |
| **Marital status** | | | | | | | 0.07 | | | | | | | - |
| Married or Common Law | 162 | 97.0 | 163 | 93.1 | 147 | 91.3 | | 324 | 100.0 | 326 | 100.0 | 325 | 100.0 | |
| Never Married | 1 | 0.6 | 9 | 5.1 | 11 | 6.8 | | 0 | 0.0 | 0 | 0.0 | 0 | 0.0 | |
| Divorced or Separated | 4 | 2.4 | 3 | 1.7 | 3 | 1.9 | | 0 | 0.0 | 0 | 0.0 | 0 | 0.0 | |
| **Smoking Hx** | | | | | | | 0.20 | | | | | | | 0.09 |
| Never smoked | 125 | 65.8 | 122 | 64.6 | 113 | 59.2 | | 324 | 100.0 | 326 | 100.0 | 319 | 98.8 | |
| Quit before this pregnancy | 33 | 17.4 | 24 | 12.7 | 31 | 16.2 | | 0 | 0.0 | 0 | 0.0 | 2 | 0.6 | |
| Quit during this pregnancy | 25 | 13.2 | 35 | 18.5 | 31 | 16.2 | | 0 | 0.0 | 0 | 0.0 | 2 | 0.6 | |
| Currently smoking | 7 | 3.7 | 8 | 4.2 | 16 | 8.4 | | - | - | - | - | - | - | |
| **Employment status** | | | | | | | <0.001 | | | | | | | <0.001 |
| Unemployed or retired | 24 | 12.4 | 25 | 12.9 | 46 | 23.7 | | 141 | 43.8 | 114 | 35.1 | 189 | 58.2 | |
| Employed part-time | 33 | 17.1 | 44 | 22.7 | 44 | 22.7 | | 30 | 9.3 | 30 | 9.2 | 28 | 8.6 | |
| Employed full-time | 136 | 70.5 | 125 | 64.4 | 104 | 53.1 | | 151 | 47.0 | 181 | 55.7 | 108 | 33.2 | |
| **Annual household income** | | | | | | | <0.001 | | | | | | | 0.06 |
| <$30K | 11 | 5.8 | 7 | 3.7 | 26 | 13.5 | | 83 | 29.9 | 67 | 23.8 | 71 | 25.4 | |
| $30K-49 999 | 15 | 7.9 | 20 | 10.5 | 31 | 16.1 | | 92 | 33.1 | 81 | 28.7 | 74 | 26.4 | |
| ≥$50K | 163 | 86.2 | 163 | 85.8 | 135 | 70.3 | | 103 | 37.1 | 134 | 47.5 | 135 | 48.2 | |
| **Mom completed high school** | 191 | 99.0 | 190 | 97.9 | 188 | 96.9 | 0.16 | 323 | 99.7 | 325 | 99.7 | 323 | 99.4 | |
| **Hours of active sport/week during pregnancy (median/IQR)** | 2 | (0–4) | 1 | (0–3) | 0.5 | (0–3) | 0.04 | 0 | (0–4) | 0 | (0–4) | 0 | (0–2) | <0.01 |
| **Social Disadvantage index** | | | | | | | 0.01 | | | | | | | 0.001 |
| Low | 141 | 86.5 | 152 | 88.4 | 118 | 73.8 | | 116 | 42.2 | 154 | 54.6 | 112 | 40.0 | |
| Moderate | 17 | 10.4 | 16 | 9.3 | 33 | 20.6 | | 122 | 44.4 | 90 | 31.9 | 111 | 39.6 | |
| High | 5 | 3.1 | 4 | 2.3 | 9 | 5.6 | | 37 | 13.5 | 38 | 13.4 | 57 | 20.4 | |
| **GDM** | 24 | 12.4 | 32 | 16.5 | 35 | 18.0 | 0.13 | 75 | 23.1 | 80 | 24.5 | 86 | 26.5 | 0.32 |
| **GDM by CDA** | 5 | 2.7 | 6 | 3.2 | 3 | 1.6 | 0.52 | 20 | 6.6 | 22 | 7.2 | 28 | 9.1 | 0.23 |
| **GDM by IADPSG** | 23 | 12.3 | 29 | 15.5 | 31 | 16.8 | 0.23 | 67 | 22.0 | 69 | 22.7 | 73 | 23.8 | 0.59 |
| **GDM by BiB** | - | | - | | - | | - | 115 | 35.4 | 113 | 34.7 | 125 | 38.5 | 0.41 |

Notes

[†]P-trend were calculated with the use of linear regression, Jonckheere-Terpstra, Cochran-Armitage or Cochran Mantel-Haenszel tests, where appropriate.

**+Data are presented as mean ± SD, median (interquartile range) or percentage, where appropriate. Percentages are rounded to nearest whole number.

GDM The mother reports having GDM or using insulin during pregnancy, it is specified on her birth chart, or her OGTT came out positive, using IADPSG thresholds.

GDM by CDA–The mother meets or exceeds at least two CDA-defined OGTT thresholds.

GDM by IADPSG–The mother meets or exceeds any of the OGTT thresholds defined by the IADPSG: Base = 5.1; 1 hour = 10.0; 2 hour = 8.5.

GDM by BiB–The mother reports having GDM or using insulin during pregnancy, it is specified on her birth chart, or her OGTT came out positive, using Born-in-Bradford thresholds.

**Table 2. Dietary factors according to categories of dietary intake of total red meat among 581 expectant mothers in the FAMILY cohort and 976 expectant mothers in the START cohort.**

| Variable | FAMILY (n = 581) | | | | | | p-trend + | START (n = 976) | | | | | | p-trend + |
|---|---|---|---|---|---|---|---|---|---|---|---|---|---|---|
| | Low | | Med | | High | | | Low | | Med | | High | | |
| | M | SD | M | SD | M | SD | | M | SD | M | SD | M | SD | |
| **Participants (n/%)** | 193 | 33.2 | 194 | 33.4 | 194 | 33.4 | - | 325 | 33.3 | 326 | 33.4 | 325 | 33.3 | - |
| **Median Total red meat (g/d)** | 30.3 | | 58.2 | | 96.7 | | | 0.06 | | 1.5 | | 20.5 | | |
| **Nutrients** | | | | | | | | | | | | | | |
| Total energy intake (kcal) | 1793 | (612) | 2113 | (640) | 2672 | (689) | <0.001 | 1779 | (613) | 1829 | (645) | 1758 | (694) | 0.68 |
| Carbohydrate (%E) | 57.3 | (6.0) | 55.2 | (5.3) | 53.9 | (5.8) | <0.001 | 58.5 | (5.2) | 57.6 | (4.9) | 55.3 | (5.2) | <0.001 |
| Protein (%E) | 15.8 | (2.7) | 15.9 | (2.3) | 16.5 | (2.2) | 0.009 | 15.0 | (2.5) | 14.9 | (2.2) | 15.6 | (2.3) | 0.001 |
| Total fat (%E) | 26.4 | (4.9) | 28.3 | (4.5) | 29.2 | (4.7) | <0.001 | 26.6 | (3.9) | 27.5 | (3.8) | 29.2 | (3.9) | <0.001 |
| Alcohol (%E) | 0.0 | (0.0–0.0) | 0.0 | (0.0–0.0) | 0.0 | (0.0–0.0) | 0.99 | 0.0 | (0.0–0.0) | 0.0 | (0.0–0.0) | 0.0 | (0.0–0.0) | <0.001 |
| Saturated Fat (%E) | 9.7 | (2.6) | 10.7 | (2.2) | 11.1 | (2.4) | <0.001 | 9.5 | (2.4) | 9.5 | (2.2) | 9.7 | (1.9) | 0.25 |
| Total protein (g/d)* | 80.2 | (13.2) | 80.4 | (11.6) | 83.0 | (11.1) | 0.02 | 76.8 | (12.2) | 76.0 | (10.9) | 79.0 | (11.1) | 0.017 |
| Animal protein (g/d)* | 47.2 | (18.8) | 58.6 | (18.2) | 79.0 | (23.8) | 0.02 | 37.7 | (21.3) | 37.5 | (18.8) | 40.3 | (20.0) | <0.001 |
| Vegetable protein (g/d)* | 23.4 | (10.7) | 24.6 | (8.8) | 30.3 | (9.1) | 0.02 | 32.7 | (11.7) | 32.9 | (13.5) | 29.4 | (13.0) | <0.001 |
| Fish protein (g/d)* | 1.7 | (0.5–4.2) | 2.0 | (0.8–4.5) | 2.8 | (0.8–5.2) | 0.41 | 0.0 | (0.0–0.0) | 0.0 | (0.0–0.8) | 1.3 | (0.4–3.7) | 0.001 |
| Total carbohydrate (g/d)* | 281.8 | (31.4) | 270.9 | (27.2) | 263.9 | (29.8) | <0.001 | 287.9 | (27.2) | 283.0 | (25.3) | 270.3 | (27.2) | <0.001 |
| Fibre (g/2000 kcal) | 18.9 | (6.0) | 16.8 | (4.5) | 15.6 | (4.0) | <0.001 | 24.9 | (5.1) | 23.0 | (5.3) | 21.0 | (4.7) | <0.001 |
| Total fibre (g/d)* | 18.9 | (6.5) | 17.2 | (4.8) | 16.4 | (4.3) | <0.001 | 25.1 | (5.4) | 23.2 | (5.9) | 20.8 | (4.9) | <0.001 |
| Soluble fibre (g/d)* | 6.5 | (2.3) | 6.1 | (2.0) | 5.9 | (1.8) | <0.001 | 9.8 | (2.2) | 9.2 | (2.3) | 8.3 | (2.0) | <0.001 |
| Insoluble fibre (g/d)* | 7.5 | (3.2) | 6.8 | (2.5) | 6.3 | (2.3) | 0.01 | 9.3 | (2.8) | 9.0 | (2.6) | 8.2 | (2.5) | <0.001 |
| Sugar (g/d)* | 12.2 | (6.6) | 11.5 | (5.8) | 9.9 | (4.6) | <0.001 | 13.7 | (5.9) | 13.2 | (5.3) | 13.2 | (4.9) | 0.18 |
| Total fat (g/d)* | 59.6 | (10.9) | 63.5 | (10.0) | 65.2 | (10.4) | <0.001 | 60.7 | (8.7) | 62.6 | (8.4) | 65.9 | (8.5) | <0.001 |
| Saturated (g/d)* | 22.3 | (5.9) | 23.9 | (4.9) | 24.2 | (5.0) | <0.001 | 21.8 | (5.4) | 21.9 | (5.0) | 22.1 | (4.3) | 0.46 |
| Monounsaturated fat (g/d)* | 21.3 | (4.6) | 23.4 | (4.5) | 24.4 | (4.3) | <0.001 | 21.6 | (3.9) | 22.8 | (3.8) | 24.7 | (4.0) | <0.001 |
| Polyunsaturated fat (g/d)* | 8.4 | (2.2) | 8.7 | (2.0) | 9.1 | (2.1) | <0.001 | 11.3 | (2.7) | 11.7 | (2.3) | 12.0 | (2.3) | 0.001 |
| Trans-Unsaturated fat (g/d) median, IQR* | 0.2 | (0.1–0.4) | 0.3 | (0.2–0.5) | 0.4 | (0.3–0.8) | <0.001 | 0.2 | (0.1–0.3) | 0.2 | (0.1–0.3) | 0.2 | (0.1–0.3) | <0.001 |
| Cholesterol (mg/d)* | 202 | (69) | 221 | (60) | 234 | (58) | <0.001 | 134 | (92) | 174 | (104) | 252 | (110) | <0.001 |
| **Glycemic Load*** | 125.5 | (21.4) | 120.2 | (17.3) | 119.4 | (20.0) | 0.003 | 118.9 | (17.1) | 119.2 | (17.3) | 116.4 | (19.4) | 0.08 |
| **Glycemic Index*** | 47.3 | (4.3) | 47.1 | (3.9) | 47.8 | (3.7) | 0.21 | 45.1 | (3.6) | 45.7 | (3.6) | 46.5 | (4.3) | <0.001 |
| **Food Groups** | | | | | | | | | | | | | | |
| Poultry (g/d) | 20.5 | (10.1–36.6) | 22.1 | (14.5–41.2) | 33.5 | (20.1–51.9) | <0.001 | 0.5 | (0.0–1.0) | 5.4 | (1.6–16.8) | 19.8 | (10.3–29.8) | <0.001 |
| Fish (g/d) | 5.6 | (1.3–18.2) | 7.1 | (2.3–15.0) | 10.0 | (2.5–0.8) | <0.001 | 0.0 | (0.0–0.0) | 0.0 | (0.0–2.6) | 4.7 | (1.3–11.3) | <0.001 |
| Seafood (g/d) | 0.9 | (0.3–2.3) | 1.2 | (0.4–2.5) | 1.2 | (0.5–2.9) | 0.01 | 0.0 | (0.0–0.1) | 0.1 | (0.0–0.3) | 0.3 | (0.1–0.9) | <0.001 |
| Organ Meat (g/d) (median (IQR)) | 0.0 | (0.0–0.0) | 0.0 | (0.0–0.0) | 0.0 | (0.0–0.0) | 0.18 | 0.0 | (0.0–0.0) | 0.0 | (0.0–0.0) | 0.0 | (0.0–0.0) | <0.001 |
| Dairy (g/d) | 690.8 | (415.8) | 740.4 | (425.4) | 849.7 | (492.4) | 0.001 | 657.7 | (382.2) | 623.9 | (351.5) | 470.4 | (298.1) | <0.001 |
| Eggs (g/d) | 15.1 | (10.8) | 18.9 | (12.9) | 24.2 | (14.2) | <0.001 | 10.3 | (27.8) | 17.7 | (20.2) | 30.3 | (26.7) | <0.001 |
| Whole Grain (g/d) | 71.1 | (52.4) | 68.2 | (53.1) | 76.9 | (48.3) | 0.27 | 78.9 | (38.6) | 76.8 | (40.5) | 62.6 | (39.2) | <0.001 |
| Refined Grain (g/d) | 110.0 | (63.9) | 126.1 | (55.7) | 176.5 | (80.2) | <0.001 | 75.7 | (38.5) | 98.2 | (50.2) | 112.9 | (77.0) | <0.001 |
| Legumes (g/d) | 17.4 | (19.9) | 18.5 | (16.3) | 21.0 | (17.3) | 0.05 | 52.6 | (32.2) | 53.2 | (34.8) | 36.7 | (26.4) | <0.001 |
| Tofu (g/d) | 0.2 | (0.1–1.1) | 0.1 | (0.0–0.4) | 0.1 | (0.0–0.4) | 0.05 | 0.3 | (0.0–3.4) | 0.2 | (0.0–3.4) | 0.0 | (0.0–0.1) | <0.001 |

*(Continued)*

**Table 2.** (Continued)

| Variable | FAMILY (n = 581) | | | | | | | START (n = 976) | | | | | | |
|---|---|---|---|---|---|---|---|---|---|---|---|---|---|---|
| | Low | | Med | | High | | p-trend + | Low | | Med | | High | | p-trend + |
| | M | SD | M | SD | M | SD | | M | SD | M | SD | M | SD | |
| Nuts and Peanuts (g/d) | 11.9 | (9.2) | 13.2 | (11.6) | 14.3 | (11.0) | 0.26 | 7.9 | (9.6) | 9.1 | (8.5) | 7.5 | (7.8) | 0.52 |
| **Diet Quality Score** | 3.6 | (5.2) | 2.5 | (5.1) | 1.5 | (6.2) | <0.001 | 9.0 | (4.6) | 8.3 | (6.0) | 6.2 | (5.2) | <0.001 |

Kcal–kilocalorie; %E–percentage of energy.

*Values presented are energy-adjusted.

†P–trend were calculated with the use of linear regression, Jonckheere-Terpstra, Cochran-Armitage or Cochran Mantel-Haenszel tests, where appropriate.

**+Data are presented as mean ± SD, median (interquartile range) or percentage, where appropriate. Percentages are rounded to nearest whole number.

only processed meat was associated with development of GDM (p-trend = 0.04), but this was attenuated with adjustment for additional dietary factors such as the diet quality score, total fiber, saturated fat, and glycemic load. The variance inflation factor did not detect multicollinearity (all values were <2 and tolerance ≥0.7) between covariates in the fully adjusted model for both START and FAMILY cohorts.

Sensitivity analysis for odds of GDM based on other diagnostic threshold (S4A–S4C Table) and consumption of red meat as a continuous variable (per 90g/d increment) (S4 Table) revealed similar results. In FAMILY, there was a trend (p = 0.05) favouring higher odds of GDM with higher processed meat consumption when the CDA definition was used to define a case, but the confidence intervals were wide and the number of cases small (n = 14).

## Discussion

In this analysis of two prospective birth cohort studies, we observed that medium consumption compared to non-consumption, low or high consumption of unprocessed red meat, processed meat, or total red meat, was not associated with increased odds of GDM. Schoenaker et al. [35] systematically reviewed observational studies of the associations between dietary factors and GDM and found that red meat increased risk of GDM by as much as 66%. However, most of the data were contributed by a single study in the USA. Reviews have mostly been on red and processed meat and the risk of T2DM, which has a similar pathogenesis as GDM. In one systematic review, Zeraatkar et al. [36] found that the strength of the evidence supporting reducing unprocessed and processed red meat intake to reduce risk of T2DM was "very low" using the GRADE approach to evaluating evidence, (14 studies, 669,530 participants; RR 0.78 [95% CI: 0.72–0.84 for reducing processed red meat by 3 servings/week and 6 studies, 293,869 participants; RR 0.90 [95% CI:0.88–0.92 for reducing unprocessed red meat by 3 servings/week). However, another review judged that there was a 'high' strength of evidence that increased consumption of red meat (14 studies, 43,781 participants; RR 1.17 [95% CI:1.08–1.26] per 100g/d) and processed meat (14 studies, 43,781 participants; RR 1.37 [95% CI:1.22–1.54] per 50g/d) increased risk of T2DM [37]. Though the approaches to evaluating the strength of the evidence differ, the risks associated with red and processed meats are similar in both papers.

Dietary patterns characterized by higher intakes of refined grains or lower intakes of fiber have independently been associated with T2DM [38,39]. Carbohydrate-rich and high glycemic index diets may increase inflammation, impair glucose homeostasis, and have been associated with increased risk of developing T2DM [38,39]; however, relatively low carbohydrate and high fat and protein intakes, particularly from animal sources, have been associated with greater risk of GDM, whereas higher fibre intake was associated with decreased risk of GDM

**Table 3. Association between low and high red and processed meat intake relative to medium intake and gestational diabetes mellitus (GDM).**

| Variable | FAMILY (n = 581) | | | | | | START (n = 976) | | | | | | | |
| --- | --- | --- | --- | --- | --- | --- | --- | --- | --- | --- | --- | --- | --- | --- |
| | Low | | Medium | High | | p-trend | NC | | Low | | Medium | High | | p-trend |
| | OR | 95% CI | | OR | 95% CI | | OR | 95% CI | OR | 95% CI | | OR | 95% CI | |
| **Unprocessed Red Meat median (g/d)** | **20.5** | | **44.1** | **75.6** | | | | | **0.06** | | **1.3** | **19.7** | | |
| GDM cases/ pregnancies | 29/193 | | 27/194 | 35/194 | | | | | 75/325 | | 81/326 | 85/325 | | |
| Unadjusted | 1.04 | 0.62–1.93 | 1 (Ref.) | 1.36 | 0.79–2.35 | 0.38 | - | - | 0.91 | 0.63–1.30 | 1 (Ref.) | 1.07 | 0.75–1.52 | 0.44 |
| Model 1 | 1.08 | 0.61–1.91 | 1 (Ref.) | 1.39 | 0.80–2.42 | 0.32 | - | - | 0.96 | 0.66–1.38 | 1 (Ref.) | 1.15 | 0.80–1.65 | 0.32 |
| Model 2 | 1.09 | 0.59–2.04 | 1 (Ref.) | 1.21 | 0.66–2.22 | 0.70 | - | - | 0.99 | 0.68–1.45 | 1 (Ref.) | 1.16 | 0.80–1.69 | 0.36 |
| Model 3 | 1.07 | 0.53–2.13 | 1 (Ref.) | 1.35 | 0.67–2.73 | 0.52 | - | - | 1.01 | 0.69–1.49 | 1 (Ref.) | 1.04 | 0.71–1.53 | 0.83 |
| Model 4 | 1.11 | 0.55–2.23 | 1 (Ref.) | 1.29 | 0.63–2.64 | 0.69 | - | - | 0.99 | 0.67–1.47 | 1 (Ref.) | 1.08 | 0.72–1.60 | 0.68 |
| *Processed Meat median (g/d) | **4.7** | | **11.5** | **24.3** | | | **0.00** | | **0.02** | | **0.22** | **1.4** | | |
| GDM cases/ pregnancies | 21/193 | | 32/194 | 38/194 | | | 129/565 | | 45/139 | | 32/135 | 35/137 | | |
| Unadjusted | 0.62 | 0.34–1.12 | 1 (Ref.) | 1.23 | 0.73–2.07 | 0.03 | 0.95 | 0.61–1.48 | 1.54 | 0.91–2.63 | 1 (Ref.) | 1.10 | 0.64–1.92 | 0.80 |
| Model 1 | 0.60 | 0.33–1.09 | 1 (Ref.) | 1.30 | 0.77–2.20 | 0.01 | 0.96 | 0.61–1.51 | 1.42 | 0.82–2.46 | 1 (Ref.) | 1.05 | 0.60–1.84 | 0.97 |
| Model 2 | 0.83 | 0.43–1.60 | 1 (Ref.) | 1.57 | 0.87–2.84 | 0.04 | 0.97 | 0.61–1.54 | 1.48 | 0.85–2.60 | 1 (Ref.) | 0.98 | 0.55–1.75 | 0.76 |
| Model 3 | 0.88 | 0.44–1.76 | 1 (Ref.) | 1.54 | 0.79–3.02 | 0.11 | 0.95 | 0.59–1.53 | 1.50 | 0.85–2.66 | 1 (Ref.) | 0.97 | 0.53–1.71 | 0.70 |
| Model 4 | 0.94 | 0.47–1.91 | 1 (Ref.) | 1.51 | 0.77–2.96 | 0.18 | 0.90 | 0.56–1.47 | 1.51 | 0.85–2.68 | 1 (Ref.) | 0.98 | 0.54–1.77 | 0.90 |
| **Total Red and Processed Meat median (g/d)** | **30.3** | | **58.2** | **96.7** | | | - | - | **0.06** | | **1.5** | **20.5** | | |
| GDM cases/ pregnancies | 24/193 | | 32/194 | 35/194 | | | - | - | 75/325 | | 80/326 | 86/325 | | |
| Unadjusted | 0.72 | 0.41–1.27 | 1 (Ref.) | 1.11 | 0.66–1.89 | 0.14 | - | - | 0.92 | 0.64–1.32 | 1 (Ref.) | 1.11 | 0.78–1.57 | 0.35 |
| Model 1 | 0.71 | 0.40–1.27 | 1 (Ref.) | 1.15 | 0.68–1.96 | 0.11 | - | - | 0.98 | 0.68–1.41 | 1 (Ref.) | 1.20 | 0.83–1.73 | 0.23 |
| Model 2 | 0.70 | 0.37–1.32 | 1 (Ref.) | 0.99 | 0.55–1.78 | 0.33 | - | - | 1.03 | 0.70–1.50 | 1 (Ref.) | 1.23 | 0.84–1.78 | 0.26 |
| Model 3 | 0.68 | 0.33–1.39 | 1 (Ref.) | 1.26 | 0.63–2.52 | 0.17 | - | - | 1.32 | 0.87–2.00 | 1 (Ref.) | 1.19 | 0.79–1.81 | 0.60 |
| Model 4 | 0.69 | 0.35–1.36 | 1 (Ref.) | 1.19 | 0.61–2.31 | 0.30 | - | - | 1.05 | 0.72–1.55 | 1 (Ref.) | 1.12 | 0.77–1.65 | 0.44 |

Model 1: Adjusted for age and parity.

Model 2: Adjusted for age, parity, pre-pregnancy BMI, pregnancy weight gain.

Model 3: Adjusted for age, parity, pre-pregnancy BMI, pregnancy weight gain, smoking (FAMILY only), family history of DM, level of education, total energy.

Model 4: Adjusted for age, parity, pre-pregnancy BMI, pregnancy weight gain, smoking (FAMILY only), family history of DM, level of education, total energy, diet quality score, total fiber, saturated fat and glycemic load.

*ORs were estimated for GDM among high, low, and non-consumers (NC) of processed meat relative to medium consumers of processed meat in START cohort.

[21,40]. Red meat may be neutral or reduce the risk of GDM because its high protein and fat content may displace highly refined carbohydrates from the diet. It is also a good source of protein, vitamin B12, and iron, which are nutrients of concern during pregnancy. In our study, White Europeans had higher intakes of refined grains, and lower intakes of fiber (based

on a 2000 kcal diet) than South Asians, but had a lower prevalence of GDM. This indicates that there are other factors (such as genetics) associated with GDM in the South Asian cohort. Lamri et al [41] in their study among pregnant South Asian women from two birth cohort studies reported that increased polygenic risk score was associated with 45% increase in risk of GDM, independent of low diet quality and parental history of T2D.

Major dietary protein sources consumed by White Europeans in our study appeared to be a mixture of animal, fish, and plant-based protein for high consumers of total red meat. For South Asian expectant mothers, we observed lower intake of poultry, fish, seafood, and refined grain (because this is a frequently consumed food group) as major protein sources. South Asians also reported consuming more vegetable protein than White Europeans, which may offset some of the proposed harms of red and processed meats. Raghavan et al. [42], in their systematic review of dietary pattern and risk of GDM among healthy White European women, reported that diets higher in vegetables, fruits, whole grains, nuts, legumes, and fish, and lower in red meat lowered risk of GDM.

South Asians in our cohort who consumed higher amounts of total red and processed meat also consumed higher protein, and carbohydrates and dietary fiber, and had a higher glycemic load when compared to lower total red and processed meat consumers. Animal models of GDM have shown that the GDM promoting effects of high-GI sucrose and maltodextrin in dams and their offspring can be reversed by substituting rapidly-metabolized carbohydrate with more slowly-digesting carbohydrate (i.e., isomaltulose and resistant maltodextrins) [43–45]. Current dietary guidelines from the International Federation of Gynecology and Obstetrics, Endocrine Society, American College of Obstetrics and Gynecologists, National Institute for Health and Care Excellence (NICE) guidelines, Diabetes Canada, American Academy of Nutrition and Dietetics, and the American Diabetes Association, recommend either to restrict carbohydrate intake or to replace higher glycemic-index carbohydrates with those that are more slowly digesting [11,46–51]. This might also prove useful in the prevention of GDM, providing healthy replacements for high-glycemic carbohydrates shown to improve glucose tolerance. It is also possible that in a population with low red meat intake, the addition of small amounts, consistent with the prevailing message of moderation espoused by most dietetic and health organizations, does not pose serious health risks, and can help ensure nutritional adequacy. In a population with typical red meat intake, increasing to higher levels may be cause for concern. Though several countries recommend the moderate consumption of meat there is much ambiguity with regards to the number of servings along with portion size which constitutes 'moderation'. Food-based dietary guidelines within Europe do offer some guidance but the recommendations vary from having 1 serving of red meat per week with a serving size of 120 – 150g (Greece) to reducing consumption to 70g/d for individuals who consume more than 90g/d red and/ or processed meat (United Kingdom) [52]. The Dietary Guidelines for Americans [53] recommends for its population consuming 26 ounces equivalent /wk for meats, poultry, eggs based on a 2,000 calorie diet. However, it is important to note that there is a running theme with the promotion of more lean cuts of red meat or processed meat. In India, there is a general recommendation of moderate consumption of meat [54]. The Canadian dietary guidelines make no reference to meat consumption, however, it recommends that the general population (2 years and older) should "choose protein foods that come from plants more often, and try to eat food with healthy fats instead of saturated fat, as well as to limit highly processed foods in small amounts." [55].

We adjusted for a measure of the overall diet quality in our models to account for variability in dietary patterns. Furthermore, the inclusion of fiber, saturated fat and glycemic load in the model address the complex mechanistic nature of insulin control. Zhou et al. [23] studied maternal dietary patterns in 2,755 Chinese women and found that a high fish-meat-eggs score

was positively related to protein intake and inversely related to carbohydrate intake. Higher scores were associated with higher odds of GDM (adjusted OR for quartile 4 v. quartile 1: 1·83; 95% CI 1·21, 2·79; p trend = 0.007) and higher plasma glucose levels. In contrast, a high rice-wheat-fruits score was positively related to carbohydrate intake and inversely related to protein intake. It was associated with lower odds of GDM (adjusted OR for quartile 3 v. quartile 1: 0·54; 95% CI 0·36, 0·83; p trend = 0·010). However, Wen et al., [56] conducted a birth cohort study of 324 twin-pregnancies in Chonqing City (n = 101 cases). Four dietary patterns were identified: a vegetable-based pattern, a poultry-and-fruit-based pattern, a sweet-based pattern and a plant-protein-based pattern. None of the dietary patterns were associated with the risk of GDM among women who were pregnant with twins. Thus, our understanding of the role of dietary patterns in the development of GDM is still evolving.

There may be other explanations for the lack of association between red and processed meat intake and GDM in our cohorts that relate to other known risk factors for GDM such as age and adiposity. Age was similar between our cohorts (FAMILY– 31 years old; START– 30 years old) and similar to other large studies [21,22,24,25,35]. On the other hand, adiposity as reflected by pre-pregnancy BMI varied between cohorts. For South Asians, mean pre-pregnancy BMI was normal (23–24 kg/m$^2$), while for White Europeans, values fell in the over-weight category (25–27 kg/m$^2$). The findings observed among the START cohort are consistent with that of Schoenacker et al. [35] among Australian women who reported greater pre-pregnancy BMI and risk of GDM. Interestingly this characteristic is similar to other studies among varying ethnic groups (NHS study [22]–mostly Caucasian and Sun Project [24]–Spanish). Higher BMI and obesity have been shown to promote insulin resistance. In our model, both higher pre-pregnancy BMI and age were associated with GDM. However, results were directionally consistent in FAMILY and not START cohort for those with BMI >25. In this group, intake of higher total red and processed meat was associated with higher odds of GDM [highest vs. medium, OR = 1.32 (95% CI: 0.0.58–5.59) for FAMILY; OR = 1.02 (95% CI: 0.55–1.89) for START; P-het = 0.69; I-squared = 0%]; however, no significant trend was seen within either cohort (S5 Table).

Our study has several strengths. We measured diet with validated semi-quantitative food-frequency questionnaires and quantified the intakes of different types of red meat. We looked at two diverse cohorts with different intakes of red meat, and GDM prevalence. As with most dietary recommendations for moderate red meat intake we used medium intake as our reference group which is more reflective contextually. Our study provides data on South Asian expectant mothers (a high-risk group for GDM) with low consumption of red and processed meat and high prevalence of GDM. Furthermore, previous studies of red and processed meat and GDM risk have not included large numbers of high-risk ethnic groups. Our South Asian birth cohort is one of the largest cohorts of pregnant women in North America, and ideally suited to assess the robustness of associations observed in White Europeans. The results indicate that it might be important to examine other factors as contributors to the odds of GDM in the population, because we have not replicated previous associations of red and processed meat with GDM risk [21,22,24]. Certainly, this observation could be due to a small number of events, or the low levels of red meat consumption by members of our South Asian cohort, but nevertheless further exploration in warranted.

We acknowledge that there are some limitations. First, our meat exposure variables relied on participants' self-report through the FFQ, an approach that is subject to recall basis, though the tools we utilized have been validated specifically for the ethnic groups in the studies, which we hope minimized the possible reporting biases due to inadequate representation of ethnic foods consumed. Second, we observed an almost perfect confounding between exposure, cohorts and outcome (i.e., all those with low/no red meat intake were from START, who have

a high risk of GDM and; all those with high red meat intake were from FAMILY, who have a lower risk of GDM. Therefore, a pooled analysis would make it difficult to separate the effects of cohort from the true effects of the dietary exposure. Thus, we kept the analyses of the cohorts separate, which is an approach we would encourage in future studies with this sort of confounding structure. Previous papers have assessed this association among populations with relatively homogeneous dietary intakes [22,24]. Third, our sample size was small, which reduced our power in multivariable analyses. Nevertheless, our point estimates in FAMILY are aligned with larger studies which have reported positive associations between red and processed meats and GDM [21,22,24]. We also had a relatively high event proportion or GDM (16%—FAMILY and 25%—START) compared to other studies which had approximately 5% [21,22,24,25]. Fourth, in both cohorts, dietary assessment and the OGTT were completed at the same visit (i.e., cross-sectionally), however at the time of FFQ administration, GDM status was not yet known to participants. Additionally, the FFQ inquired about food consumed in the previous 1-year/duration thus, associations are prospectively determined. Finally, the cohorts represent two time periods, roughly 7 years apart. Red (pork, sheep, veal and beef) consumption per capita in Canada between 2004 to 2009 (44.69 kg– 39.93 kg) has decreased when compared to 2011 to 2015 (37.48 kg– 35.54) [57].

## Conclusion

To our knowledge, our study is the first to examine the association of red and processed meat and GDM in South Asian population in a developed country. The results indicate that medium compared with low or high red and processed meat intakes were not associated with GDM among White Europeans or South Asians (despite the high percentage of non-consumers and low consumers) living in Canada. Future studies are needed to understand the role of modifiable and non-modifiable factors that influence the relationship between dietary protein intake and GDM.

## Supporting information

**S1 Fig. Sample population CONSORT flowchart.**
(DOCX)

**S1 Table. Listing of red meat and processed meat classification by cohorts.**
(DOCX)

**S2 Table. Participant characteristics according to categories of dietary intake of red meat and processed red meat among 581 expectant mothers in the FAMILY cohort and 976 expectant mothers in the START cohort.**
(DOCX)

**S3 Table.** A. Association between low and high red and processed meat intake relative to medium intake and gestational diabetes mellitus (GDM)–CDA. B. Association between low and high red and processed meat intake relative to medium intake and gestational diabetes mellitus (GDM)–IADPSG. C. Association between low and high red and processed meat intake relative to medium intake and gestational diabetes mellitus (GDM)–Born-in-Bradford.
(DOCX)

**S4 Table. Odds Risk for gestational diabetes mellitus (GDM) according to 90 grams of red and processed meat intake by cohort.**
(DOCX)

**S5 Table.** A. Odds Ratio (95% confidence interval) for gestational diabetes mellitus (GDM) according to red and processed meat intake stratified by BMI–FAMILY cohort. B. Odds Ratio (95% confidence interval) for gestational diabetes mellitus (GDM) according to red and processed meat intake stratified by BMI–START cohort.
(DOCX)

## Acknowledgments

We thank the participants of the START and FAMILY studies for their participation and dedication to our research. START Canada Investigators: Sonia S. Anand, Anil Vasudevan, Milan Gupta, Katherine Morrison, A Kurpad, Koon K. Teo and Krishnaramachari Srinivasan. START Research Personnel: Nora Abdalla, Nilofur Aga, Dipika Desai, Shariar Hirjikaka, Manpreet Kooner, Farah Khan, Monisha Nundy, Zahra Sohani, Natalie Williams and Sherry Zafar. This work was supported by Canadian Institutes of Health Research (Grant INC-109205), and the Heart and Stroke Foundation of Canada: (Grant NA 7283). We also thank FAMILY Study Investigators: Katherine Morrison, Stephanie Atkinson, Koon K. Teo, Jacqueline Bourgeois, Jan W. Jansen, Matthew McQueen, Salim Yusuf; FAMILY Staff, (Population Health Research Institute, Hamilton Health Sciences and McMaster University, Hamilton, Canada), FAMILY Research Personnel: Nora Abdalla, Dipika Desai, Julie Gross, Liz Helden, Natalie Jakymyshyn, Sumaira Khurshid, Carol McLean-Price, Sumathy Rangarajan, Purnima Rao Melacini, Lynn Rusnak, Karleen Schulze, Cathy Sim, Susan Steele, Aisha Van Der Loo, Vivian Vaughan Williams and Catherine Wright.

## Author Contributions

**Conceptualization:** Rosain N. Stennett, Hertzel C. Gerstein, Shrikant I. Bangdiwala, Sonia S. Anand, Russell J. de Souza.

**Data curation:** Rosain N. Stennett, Russell J. de Souza.

**Formal analysis:** Rosain N. Stennett, Shrikant I. Bangdiwala, Talha Rafiq, Sonia S. Anand, Russell J. de Souza.

**Funding acquisition:** Koon K. Teo, Katherine M. Morrison, Stephanie A. Atkinson, Sonia S. Anand.

**Methodology:** Rosain N. Stennett, Hertzel C. Gerstein, Shrikant I. Bangdiwala, Sonia S. Anand, Russell J. de Souza.

**Supervision:** Hertzel C. Gerstein, Shrikant I. Bangdiwala, Sonia S. Anand, Russell J. de Souza.

**Writing – original draft:** Rosain N. Stennett, Hertzel C. Gerstein, Shrikant I. Bangdiwala, Talha Rafiq, Koon K. Teo, Katherine M. Morrison, Stephanie A. Atkinson, Sonia S. Anand, Russell J. de Souza.

**Writing – review & editing:** Rosain N. Stennett, Hertzel C. Gerstein, Shrikant I. Bangdiwala, Talha Rafiq, Koon K. Teo, Katherine M. Morrison, Stephanie A. Atkinson, Sonia S. Anand, Russell J. de Souza.

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
