## [Decision Letter · Decision Letter 0]

20 Feb 2023

PONE-D-22-22041The association of red meat with gestational diabetes mellitus: results from 2 Canadian birth cohort studiesPLOS ONE

Dear Dr. de Souza,

Thank you for submitting your manuscript to PLOS ONE. After careful consideration, we feel that it has merit but does not fully meet PLOS ONE’s publication criteria as it currently stands. Therefore, we invite you to submit a revised version of the manuscript that addresses the points raised during the review process.

We look forward to receiving your revised manuscript.

Kind regards,

George Kuryan

Academic Editor

PLOS ONE

Journal Requirements:

2. Thank you for providing the following Funding Statement: 

 “I have read the journal's policy and the authors of this manuscript have the following competing interests: RJ de Souza has served as an external resource person to the World Health Organization’s Nutrition Guidelines Advisory Group on trans fats, saturated fats, and polyunsaturated fats. The WHO paid for his travel and accommodation to attend meetings from 2012-2017 to present and discuss this work. He has presented updates of this work to the WHO in 2022. He has also done contract research for the Canadian Institutes of Health Research’s Institute of Nutrition, Metabolism, and Diabetes, Health Canada, and the World Health Organization for which he received remuneration. He has received speaker’s fees from the University of Toronto, and McMaster Children’s Hospital. He has held grants from the Canadian Institutes of Health Research, Canadian Foundation for Dietetic Research, Population Health Research Institute, and Hamilton Health Sciences Corporation as a principal investigator, and is a co-investigator on several funded team grants from the Canadian Institutes of Health Research. He has served as an independent director of the Helderleigh Foundation (Canada). He serves as a member of the Nutrition Science Advisory Committee to Health Canada (Government of Canada), and a co-opted member of the Scientific Advisory Committee on Nutrition (SACN) Subgroup on the Framework for the Evaluation of Evidence (Public Health England).”

We note that one or more of the authors is affiliated with the funding organization, indicating the funder may have had some role in the design, data collection, analysis or preparation of your manuscript for publication; in other words, the funder played an indirect role through the participation of the co-authors.

If the funding organization did not play a role in the study design, data collection and analysis, decision to publish, or preparation of the manuscript and only provided financial support in the form of authors' salaries and/or research materials, please review your statements relating to the author contributions, and ensure you have specifically and accurately indicated the role(s) that these authors had in your study in the Author Contributions section of the online submission form. Please make any necessary amendments directly within this section of the online submission form.  Please also update your Funding Statement to include the following statement: “The funder provided support in the form of salaries for authors [insert relevant initials], but did not have any additional role in the study design, data collection and analysis, decision to publish, or preparation of the manuscript. The specific roles of these authors are articulated in the ‘author contributions’ section.”

If the funding organization did have an additional role, please state and explain that role within your Funding Statement.

Please also provide an updated Competing Interests Statement declaring this commercial affiliation along with any other relevant declarations relating to employment, consultancy, patents, products in development, or marketed products, etc. 

3. PLOS requires an ORCID iD for the corresponding author in Editorial Manager on papers submitted after December 6th, 2016. Please ensure that you have an ORCID iD and that it is validated in Editorial Manager. To do this, go to ‘Update my Information’ (in the upper left-hand corner of the main menu), and click on the Fetch/Validate link next to the ORCID field. This will take you to the ORCID site and allow you to create a new iD or authenticate a pre-existing iD in Editorial Manager. Please see the following video for instructions on linking an ORCID iD to your Editorial Manager account: https://www.youtube.com/watch?v=_xcclfuvtxQ"

Additional Editor Comments (if provided):

concur with the reviewer.

This study investigates the association between red meat intake and gestational diabetes mellitus (GDM) risk in two individual cohorts. However, no significant associations were observed in both cohorts. Some improvements can be made before being considered for publication, I list some suggestions and concerns below:

Introduction

1. Please include some background information on the meat consumption pattern in Canada and other related regions

2. Line 97-99: “We conducted a prospective … age 18 – 40 years old.” Please revisit this statement as this study did not conduct an actual prospective study

Methodology - The methods section requires more details to ensure clarity

1. Please justify the rationale for using the data from 2 cohorts. If I understand correctly from the methodology section, the FAMILY cohort consisted of multiple ethnic groups, including the South East Asian and South Asian. Please justify why only White Europeans were selected from FAMILY, why this study only included the findings from the FAMILY cohort as it consisted of multiple ethnic groups – and further comparison in the associations between red meat intake and GDM among ethnic groups can be made. Besides, there is a time difference between START (2011-2015) and FAMILY cohorts (2004-2009), whereby the eating patterns were somehow different.

2. Supplement figure 1: Please give more clear description regarding the flow of participants

(Consort diagram), some parts of the box cannot be read.

3. Line 196: what type of intake? Total energy intake? Red meat intake?

4. I am wondering why the authors use logistic regression for the cohort study since relative risk (RR), and hazard ratio (HR) are more suitable for conducting statistical analysis. Furthermore, previous studies (lines 85 – 91 & lines 277 - 285) also used RR to perform the analysis. Suggest setting the “medium” group as the reference group. Please confirm and change the statistical analysis.

5. I am confused about how the tertiles (low, medium and high) are determined. Detailed grouping information needs to be provided (e.g., the cut-off for each group). Were those in the low red meat group indeed had low red meat intakes compared to the previous study or standard reference?

6. Line 208 – 216: Further clarification of the multivariate logistic regression is also required. Please explain how the sequence of covariates of the model was made

Results

1. Tables: Please refer to the journal guideline on the presentation of tables.

2. Table 1: Please include n (%) for each category.

3. Table 2: Diet index (last row). Diet quality index (please use a standardized term throughout the entire manuscript)? If not, please include a description in the Methodology section

4. Suggest combining paragraphs 1 and 2 of the results and simplifying it

5. Line 240 – 247: Please revisit this paragraph, as it is confusing.

6. Line 248-259: Please revisit this paragraph, as it did not reflect all the findings in Table 2

Discussion

1. Line 270-272: This finding was not shown in the result section.

2. Line 295-299: Could the authors comment on why white Europeans had higher intakes of refined grains, and lower intakes of fiber (based on 299 a 2000 kcal diet) than South Asians, but had a lower prevalence of GDM?

3. Line 324: what is the cut-off for moderation spoused by most dietetic and health organizations, and are any references for

Reviewers' comments:

Reviewer's Responses to Questions

**Comments to the Author**

1. Is the manuscript technically sound, and do the data support the conclusions?

Reviewer #1: Yes

2. Has the statistical analysis been performed appropriately and rigorously? 

Reviewer #1: Yes

3. Have the authors made all data underlying the findings in their manuscript fully available?

Reviewer #1: No

4. Is the manuscript presented in an intelligible fashion and written in standard English?

Reviewer #1: Yes

5. Review Comments to the Author

Reviewer #1: This study investigates the association between red meat intake and gestational diabetes mellitus (GDM) risk in two individual cohorts. However, no significant associations were observed in both cohorts. Some improvements can be made before being considered for publication, I list some suggestions and concerns below:

Introduction

1. Please include some background information on the meat consumption pattern in Canada and other related regions

2. Line 97-99: “We conducted a prospective … age 18 – 40 years old.” Please revisit this statement as this study did not conduct an actual prospective study

Methodology - The methods section requires more details to ensure clarity

1. Please justify the rationale for using the data from 2 cohorts. If I understand correctly from the methodology section, the FAMILY cohort consisted of multiple ethnic groups, including the South East Asian and South Asian. Please justify why only White Europeans were selected from FAMILY, why this study only included the findings from the FAMILY cohort as it consisted of multiple ethnic groups – and further comparison in the associations between red meat intake and GDM among ethnic groups can be made. Besides, there is a time difference between START (2011-2015) and FAMILY cohorts (2004-2009), whereby the eating patterns were somehow different.

2. Supplement figure 1: Please give more clear description regarding the flow of participants

(Consort diagram), some parts of the box cannot be read.

3. Line 196: what type of intake? Total energy intake? Red meat intake?

4. I am wondering why the authors use logistic regression for the cohort study since relative risk (RR), and hazard ratio (HR) are more suitable for conducting statistical analysis. Furthermore, previous studies (lines 85 – 91 & lines 277 - 285) also used RR to perform the analysis. Suggest setting the “medium” group as the reference group. Please confirm and change the statistical analysis.

5. I am confused about how the tertiles (low, medium and high) are determined. Detailed grouping information needs to be provided (e.g., the cut-off for each group). Were those in the low red meat group indeed had low red meat intakes compared to the previous study or standard reference?

6. Line 208 – 216: Further clarification of the multivariate logistic regression is also required. Please explain how the sequence of covariates of the model was made

Results

1. Tables: Please refer to the journal guideline on the presentation of tables.

2. Table 1: Please include n (%) for each category.

3. Table 2: Diet index (last row). Diet quality index (please use a standardized term throughout the entire manuscript)? If not, please include a description in the Methodology section

4. Suggest combining paragraphs 1 and 2 of the results and simplifying it

5. Line 240 – 247: Please revisit this paragraph, as it is confusing.

6. Line 248-259: Please revisit this paragraph, as it did not reflect all the findings in Table 2

Discussion

1. Line 270-272: This finding was not shown in the result section.

2. Line 295-299: Could the authors comment on why white Europeans had higher intakes of refined grains, and lower intakes of fiber (based on 299 a 2000 kcal diet) than South Asians, but had a lower prevalence of GDM?

3. Line 324: what is the cut-off for moderation spoused by most dietetic and health organizations, and are any references for this?

6. PLOS authors have the option to publish the peer review history of their article (what does this mean?). If published, this will include your full peer review and any attached files.

Reviewer #1: No

---

## [Author Response · Author response to Decision Letter 0]

25 Jul 2023

Introduction 

Comment 1. Please include some background information on the meat consumption pattern in Canada and other related regions 

Response 1: Thank you for your comment, we have amended the introduction section of the main paper as follows:

“This is of keen interest as national cross-sectional survey among select North American countries (US, Canada and Mexico) showed that the median (IQR) grams of unprocessed red meat was higher in Canada (79.0 (36.6–131.6) g), than in the US (72.3 (38.3–124.5) g, p = 0.04) (Frank et al. 2021). Furthermore Canadians (79.0 (36.2–140.1)g) consume higher median intakes of total meat when compared to Mexico (62.5 (31.3–117.4) g, p = 0.04), but not different from median intake in the US (79.4 (40.8–134.7) g). Page 6, Lines 114 – 120

Comment 2. Line 97-99: “We conducted a prospective … age 18 – 40 years old.” Please revisit this statement as this study did not conduct an actual prospective study 

Response 2: Thank you for this comment, we have clarified the design; the sentence now reads: 

“We explored the association of maternal red and processed meat intake with GDM in two Canadian cohorts of single gestation pregnant women (South Asian and White European) age 18 – 40 years old.” Page 6 Lines 121 – 123

Methodology - The methods section requires more details to ensure clarity 

Comment 1. Please justify the rationale for using the data from 2 cohorts. If I understand correctly from the methodology section, the FAMILY cohort consisted of multiple ethnic groups, including the South East Asian and South Asian. Please justify why only White Europeans were selected from FAMILY, why this study only included the findings from the FAMILY cohort as it consisted of multiple ethnic groups – and further comparison in the associations between red meat intake and GDM among ethnic groups can be made. Besides, there is a time difference between START (2011-2015) and FAMILY cohorts (2004-2009), whereby the eating patterns were somehow different. 

Response 1: Thank you for this comment. Our methods section has been modified and now reads: 

“We opted to use data from these two cohort studies because they represent two different ethnic groups, with different background risks for GDM, and very different dietary patterns. The START cohort includes South Asian Canadians, the largest non-white ethnic group in Canada and FAMILY cohort has data on White Europeans, the largest ethnic group in Canada. It is important to note that though the FAMILY cohort is multi-ethnic. The majority of participants (>70%) are White European. Additionally, both cohort studied assessed diet using a comparable instrument (a validated semi-quantitative food-frequency questionnaire), which improves the comparability of dietary data. Finally, these two cohorts encompassed a wide range of meat intake, which we believed would help us to model the associations more finely.” Page 7, Lines 141 – 150

Thank you, it’s a very interesting observation with regards to difference in data collection period. We have mentioned this in the discussion as a limitation. 

“Finally, the cohorts represent two time period, roughly 7 years apart. Red (pork, sheep, veal and beef) consumption per capita in Canada between 2004 to 2009 (44.69 kg – 39.93 kg) has decreased when compared to 2011 to 2015 (37.48 kg – 35.54) [48].” Page 25, Lines 406 – 409

Comment 2. Supplement figure 1: Please give more clear description regarding the flow of participants Consort diagram), some parts of the box cannot be read. 

Response 2: Thank you for this comment. We have listed the starting population for each cohort, and delineated the reasons why participants from each cohort were not included in the present analyses. 

We have modified S1 Fig. Sample Population CONSORT Flowchart (see below). 

Comment 3. Line 196: what type of intake? Total energy intake? Red meat intake? 

Response 3. Thank you we, have modified the sentence to say: 

“START participants had >8-fold lesser mean/ median red and processed meat intake than FAMILY participants; and START had 1.6-fold higher incidence of GDM than FAMILY.” Page 10, Lines 229 – 230

Comment 4. I am wondering why the authors use logistic regression for the cohort study since relative risk (RR), and hazard ratio (HR) are more suitable for conducting statistical analysis.

Furthermore, previous studies (lines 85 – 91 & lines 277 - 285) also used RR to perform the analysis. Suggest setting the “medium” group as the reference group. Please confirm and change the statistical analysis. 

Response 4: Thank you, we utilized cross-sectional data from two prospective cohort study, as a result the reporting of odds ratio was more appropriate, we apologize for this confusion.

We have modified the methods section of our paper to that:

“We used cross-sectional data from two prospective cohort studies…”. Page 6, Line 126

Additionally, the nature of cross-sectional data for this type of assessment is mentioned in the discussion – limitations.

“Fourth, in both cohorts, dietary assessment and the OGTT were completed at the same visit (i.e., cross-sectionally), however at the time of FFQ administration, GDM status was not yet known to participants. Additionally, the FFQ inquired about food consumed in the previous 1-year/duration thus, associations are prospectively determined.” Page 25, Lines 402 – 406

We have modified our analysis to now have medium intake as the reference group, consequently the following sections of the paper has been modified (presented in detail below):

1. Abstract – Results and Conclusion

2. Main manuscript

a. Methods

b. Results

c. Discussion

Comment 5. I am confused about how the tertiles (low, medium and high) are determined. Detailed grouping information needs to be provided (e.g., the cut-off for each group). Were those in the low red meat group indeed had low red meat intakes compared to the previous study or standard reference? 

Response 5. Thank you for your comment, we have provided a definition for tertile within the document which reads:

“Participants in each cohort were placed into three equally-size groups according to cohort-specific tertile cut-offs (refers to any of two points of the ordered distribution of consumers which divides the group into three equal parts) for consumption of red and processed meat (low, medium, and high).” Page 10, Lines 232 – 235

Comment 6. Line 208 – 216: Further clarification of the multivariate logistic regression is also required. Please explain how the sequence of covariates of the model was made. 

Response 6. Thank you for this comment, we have sought to explain the how the models were made through the following sentences in the document:

“Covariates entered in the models were chosen based on our reading of previous literature and included both biological and lifestyle / behavioural risk factors for GDM. To build our model, we first used a series of univariate correlation analyses testing variables known or suspected to be associated with either red meat consumption or GDM (e.g. age, prepregnancy BMI, family history of DM, etc.). Any variables associated with either exposure our outcome with a univariate correlation p<0.20 were considered for inclusion. Then, we present models sequentially adjusted for more variables, entered as sets representing an underlying concept (e.g., non-modifiable risk factors; body weight; lifestyle factors; and dietary variables).” Pages 10 - 11, Lines 246 – 254

Results 

Comment 1. Tables: Please refer to the journal guideline on the presentation of tables. 

Response 1. Thank you for this comment, we have included all tables within the main document directly after the paragraph in which it is first cited.

Pages 12-14 – Table 1 

Pages 15-17 – Table 2

Page 19 – Table 3

Comment 2. Table 1: Please include n (%) for each category. 

Response 2. Thank you for highlighting this, Table 1 has been modified to include n (%) for each category. Page 12

Comment 3. Table 2: Diet index (last row). Diet quality index (please use a standardized term throughout the entire manuscript)? If not, please include a description in the Methodology section 

Response 3. Thank you for this observation we speak about both diet quality and diet index (which is a researcher derived index score from a previous study) we have now modified the methods section of the main document to state: 

“To describe overall diet quality, we used a previously developed diet quality score (DQS), calculated as the sum of daily servings of “healthy” foods (fermented dairy, fish and seafood, leafy green vegetables, cruciferous vegetables, legumes, fruits, nuts, and whole grains) less the sum of daily servings of “unhealthy foods” (processed meats, refined grains, French fries, snacks, sweets, and sweet drinks).”

Pages 8 - 9, Lines 192 – 196

Comment 4. Suggest combining paragraphs 1 and 2 of the results and simplifying it 

Response 4. Thank you for this suggestion, we have combined paragraphs 1 and 2, and also made modified overall.

“Tables 1 and 2 show the participants’ demographic characteristics and dietary factors by intake of total red meat, separately for START and FAMILY (see Supplemental 3 for unprocessed red meat and processed meat). The median intake of unprocessed red meat was 1.34 g/d (0.2-11.0 g/d); of processed meat 0.0 g/d (0.0-0.2 g/d); and of total red and processed meat 1.5 g/d (0.2-11.3 g/d) in START (n=976). The median intake of unprocessed red meat was 44.1 g/d (27.5-65.9 g/d); of processed meat was 11.5 g/d (6.2-19.7 g/d); and of total red and processed meat was 58.2 g/d (37.1-85.7 g/d) in FAMILY (n=581). 24.7% (n = 241) of START participants and 15.7% (n = 91) of FAMILY participants were diagnosed with GDM (Supplemental 3). FAMILY participants when compared to START participants were younger and likely to smoke (currently or previously). Within the FAMILY cohort, high total red and processed meat consumers were less likely to be married or in a common-law relationship, and had lower levels of education, employment, and physical activity. START participants who had high total red and processed meat intake were more likely to be unemployed or retired, and have an annual household income >$50,000 compared to their low total red and processed meat consumer counterparts. In both cohorts, high total red and processed meat consumers had a higher pre-pregnancy BMI, and a higher SDI. Higher total red and processed meat consumers were more likely to have a family history of T2D in START.” Page11, Lines 263 - 279

Comment 5. Line 240 – 247: Please revisit this paragraph, as it is confusing. 

Response 5. We have modified the paragraph and it now reads:

“Red and processed meat consumption was associated with other nutrients and food groups that may influence cardiometabolic health. For example, high total red and processed meat consumers in START consumed higher amounts of total protein, total fat, cholesterol, and less total fibre (Table 2). High total red and processed meat consumers in FAMILY consumed higher total energy, total protein, total fat, saturated fat, cholesterol, and less total carbohydrates, total fibre, and sugar compared to low consumers. In START, higher consumers of total red and processed meat ate more poultry, fish, seafood, and eggs; and less dairy and legumes. In FAMILY, higher consumers of total red meat ate more poultry, fish, seafood, dairy, eggs, and legumes; but lower amounts of nuts.” Page 18, Lines 239 – 247

Comment 6. Line 248-259: Please revisit this paragraph, as it did not reflect all the findings in Table 2 

Response 6. Thank for your comment section of the main document has been modified as follows:

“In START, neither lower nor higher intakes of unprocessed red meat, processed meat, or total red and processed meat were associated with increased risk of GDM, when compared with medium intakes (Table 3), with no evidence of a trend across categories. In FAMILY, only processed meat was associated with development of GDM (p-trend = 0.04), but this was attenuated with adjustment for additional dietary factors such as the diet quality score, total fiber, saturated fat, and glycemic load. The test for multicollinearity did not detect any significant correlation (all variance inflation factor values were <2 and tolerance >0.7) between covariates in the fully adjusted model for both START and FAMILY cohorts.” Page 18, Lines 248 - 255

Discussion 

Comment 1. Line 270-272: This finding was not shown in the result section. 

Response 1. Thank you for your comment. The sentence has been removed from the manuscript.

Comment 2. Line 295-299: Could the authors comment on why white Europeans had higher intakes of refined grains, and lower intakes of fiber (based on 299 a 2000 kcal diet) than South Asians, but had a lower prevalence of GDM? 

Comment 2. Thank you that’s an interesting question. We have included a sentence which makes reference to the potential impact genetics.

“This indicates that they are other factors (such as genetics) associated with GDM in the South Asian cohort. Lamri [32] studied pregnant South Asian women from two birth cohort studies, and reported that increased polygenic risk score was associated with 45% increased risk of GDM, with adjustment for low diet quality and parental history of T2D.” Page 22, Lines 299 – 303

Comment 3. Line 324: what is the cut-off for moderation spoused by most dietetic and health organizations, and are any references for this? 

Response 3. Thank you for this query ww have modified the manuscript.

“Though several countries recommend the moderate consumption of meat there is much ambiguity with regards to the number of servings along with portion size which constitutes ‘moderation’. Food-based dietary guidelines within Europe do offer some guidance but the recommendations vary from having 1 serving of red meat per week with a serving size of 120 – 150g (Greece) to reducing consumption to 70g/d for individuals who consume more than 90g/d red and/ or processed meat (United Kingdom) [43]. The Dietary Guideline for the Americas [44] recommends for its population consuming 26 ounces equivalent /wk for meats, poultry, eggs based on a 2,000 calorie diet. However, it is important to note that there is a running theme with the promotion of more lean cuts of red meat or processed meat. In India, there is a general recommendation of moderate consumption of meat [45]. The Canadian dietary guidelines make no reference to meat consumption, however, it recommends that the general population (2 years and older) should “choose protein foods that come from plants more often, and try to eat food with healthy fats instead of saturated fat, as well as to limit highly processed foods in small amounts.” [46].” Page 23, Lines 330 - 343

---

## [Decision Letter · Decision Letter 1]

13 Sep 2023

PONE-D-22-22041R1The association of red and processed meat with gestational diabetes mellitus: results from 2 Canadian birth cohort studiesPLOS ONE

Dear Dr. De Souza 

Thank you for submitting your manuscript to PLOS ONE. After careful consideration, we feel that it has merit but does not fully meet PLOS ONE’s publication criteria as it currently stands. Therefore, we invite you to submit a revised version of the manuscript that addresses the points raised during the review process.

We look forward to receiving your revised manuscript.

Kind regards,

George Kuryan

Academic Editor

PLOS ONE

Journal Requirements:

Reviewers' comments:

Reviewer's Responses to Questions

**Comments to the Author**

1. If the authors have adequately addressed your comments raised in a previous round of review and you feel that this manuscript is now acceptable for publication, you may indicate that here to bypass the “Comments to the Author” section, enter your conflict of interest statement in the “Confidential to Editor” section, and submit your "Accept" recommendation.

Reviewer #2: All comments have been addressed

2. Is the manuscript technically sound, and do the data support the conclusions?

Reviewer #2: Partly

3. Has the statistical analysis been performed appropriately and rigorously? 

Reviewer #2: No

4. Have the authors made all data underlying the findings in their manuscript fully available?

Reviewer #2: No

5. Is the manuscript presented in an intelligible fashion and written in standard English?

Reviewer #2: Yes

6. Review Comments to the Author

Reviewer #2: i have mentioned all my comments in the attached comments document

I haven't reviewed this manuscript earlier. i have read the response to the comments posted by the other reviewers and authors have tried to respond.

I have concerns about tertile cutoffs for the meat intake, cohort data analysis, FFQ used in the 2 cohorts. i have mentioned all that in the attachment

7. PLOS authors have the option to publish the peer review history of their article (what does this mean?). If published, this will include your full peer review and any attached files.

Reviewer #2: No

---

## [Author Response · Author response to Decision Letter 1]

21 Feb 2024

Comment 1

Introduction 

In the introduction, I am not clear why they chose these 2 Canadian cohorts from different ethnic groups. I would suggest that the authors should build up the case why they chose the 2 cohorts with references (several previous studies are there), highlighting most number of studies done in US, association of consumption of PM, UM with GDM is not very clear in Asian population with reference etc. The authors have rightly mentioned that the evidence on the association between dietary animal protein sources and GDM is inconsistent, but they haven’t mentioned whether there were any methodological flaws/deficiencies in the previous studies and that they have addressed in this analysis to add to the evidence. 

Response 1

Thank you for your comment, we have amended the introduction and discussion sections of the main paper as follows:

Introduction

Ethnicity and diet are risk factors for GDM [10–17]. Women of African, Arab, Asian, Hispanic, Indigenous, Native American, and South Asian ethnicity are at increased risk of developing GDM. Farrar et al. [18], in the UK-based “Born-in-Bradford” birth cohort study among Europeans and South Asian women, found that the prevalence of GDM among South Asians (range 4.1 – 17.4%) was approximately twice that of White British counterparts (range: 1.2 – 8.7%), irrespective of the diagnostic cut-off threshold used. Anand et al. [19] reported a 36.3% prevalence of GDM among 1006 expectant South Asian mothers in Canada using the ethnic-specific Born-in-Bradford (BiB) criteria. However, there is uncertainty about the exact role/mechanism of some risk factors such as ethnicity in the development of GDM. Page 5, lines 101 – 110

Discussion

Furthermore, previous studies of red and processed meat and GDM risk have not included large numbers of high-risk ethnic groups. Our South Asian birth cohort is one the largest cohorts of pregnant women in North America, and ideally suited to assess the robustness of associations observed in White Europeans. Page 25, lines 382 – 385

Comment2

Methodology 

Line 127-128: We used cross-sectional data from two prospective cohort studies: the SouTh Asian birth cohorT (START) .. FAMILY

Have they used cross sectional data? What I understand from the paper is that they have used prospective cohort data, measured diet data of 12 months before pregnancy and followed up incident GDM and co-variates measured in the 2nd trimester. Analysed retrospectively. 

Response 2

Thank you for this comment. Though both birth cohorts are prospective (FAMILY and START) participants completed the food-frequency questionnaire at the same visit at which they took the OGTT (i.e., at a cross-section in time), thus the analysis is not a “clean” prospective design. We agree, participants are recalling diet prior to the OGTT results, however, to be true to the data collection approach, we have chosen to present the more conservative analysis that reflects this. We have stated this in the methods section: 

 “We used cross-sectional data from two prospective cohort studies…”. Page 6, line 136

Additionally, the nature of cross-sectional data for this type of assessment in our study is mentioned in the discussion – limitations section.

“Fourth, in both cohorts, dietary assessment and the OGTT were completed at the same visit (i.e., cross-sectionally), however at the time of FFQ administration, GDM status was not yet known to participants. Additionally, the FFQ inquired about food consumed in the previous 1-year/duration thus, associations are prospectively determined.” Page 26, lines 406 – 410

Comment 3

line 129-131, I suggest that the overall objective of the START cohort study is mentioned like they have discussed about FAMILY study in lines 136-137 

Response 3

Thank you for this comment. We have included the following sentence:

The study seeks to understand the role of fetal programming in the development of adiposity in South Asians in general - a high risk group for T2DM and CVD. Pages 6 - 7, lines 140 - 141

Comment 4

Line 148-149, I suggest that authors mention which validated semi-quantitative food-frequency questionnaire was used in these 2 cohort studies, (was SHARE FFQ in both studies?) how many items, any difference in method of analyses

Response 4

Thank you, we have modified the methods section of the paper to say: 

“The Study of Health and Risk in Ethnic Groups Food Frequency Questionnaires (SHARE-FFQs) was administered at the initial visit in both cohort studies (second to third trimester). The validated semi-quantitative 163-item SHARE-FFQ was developed for South Asians - START (163-item) and White Europeans - FAMILY (157-item) living in Canada. [29]. Briefly, for FFQs participants are asked to indicate their frequency of consumption of each item over the past 12-months (per day, per week, per month, per year, or never), with three reference serving sizes provided. A picture or common measure was shown for a medium, or typical serving, and a small serving was half the size of a medium, and a large was 1.5 to 2-times the size of a medium [29]. Reproducibility and validity of the SHARE-FFQ can be found in other publications [29–31].” Page 8, lines 183 – 193

Additionally, we have also moved the following sentence to the end of the paragraph:

“Nutrient intake obtained from the SHARE-FFQs was calculated using the Food Processor nutrient analysis software (version 6.11, 1996, ESHA, Salem, OR), derived from the 1991 Canadian Nutrient File (CFN) and the US Department of Agriculture nutrient food composition databases. The updated 1997 CFN was used to ascertain meat and poultry nutrient values. Recipes were generated for red and processed meat composite food items that were not available in the database, and a nutrient value/ profile was developed. Because serving size varies across food items (e.g., a serving of a hotdog or lunch meat is smaller than a serving of steak or meatloaf) and some items (e.g., ground beef as hamburger, meatloaf, in casseroles) are not entirely meat, the estimated average grams of meat per serving for each food item was calculated and standardized by a registered dietitian familiar with South Asian and European food items. Grams of meat for each sub-item were totalled to get an estimate for each composite item (e.g., baked ham is 100% meat; ground beef as hamburger, meatloaf, in casseroles is 84% meat).” Page 8, lines 194 - 205

Comment 5

Line 150-151 Though the authors argued that the ‘two cohorts encompassed a wide range of meat intake, which we believed would help model the associations more finely’, they finally decided to analyse separately the data from 2 cohorts(decided to examine the data based on cohort-specific intake of red and processed meat) as they found START cohort with lower meat consumption reported higher prevalence of GDM. Didn’t the sample size have sufficient power to compare the association between different major ethnic groups and tease out the major confounders, effect modifiers to make this paper more valuable? 

Response 5

Thank you for your comment we have included the following sentences in the methods section:

“Additionally, both cohort studies assessed diet using a comparable instrument (SHARE FFQs - a validated semi-quantitative food-frequency questionnaires), which improved the comparability of dietary data. To account for the intractable association/ confounding with red and processed meat intake and ethnicity (i.e., cohort) we created cohort-specific groups.” Page 7, lines 162 - 165

Regarding sample size and power calculations, we did not intend to compare the direct association of red and processed meat intake in the original birth cohort studies. Indeed, given our fixed sample size and number of cases we do not believe that a larger sample size would have been more robust in our analysis. Our results are comparable to that of the Nurse’s Health Study 2 by Zhang et al. (2006) (a larger cohort study with 758 GDM cases of 13,110 participants). 

In our power calculation for this study, we considered previous studies on the association between red and processed meat and GDM specifically we were guided by data in the Zhang et al. (2006). In this study higher compared with lower intake of red and processed meat were assessed for their association with GDM. These associations were assessed on the background of western and prudent dietary pattern. The western dietary pattern is very similar to that of the participants in FAMILY and the prudent is similar to START. The contrast between high and low for the western group was 88.4 (total red + processed meat), while the contrast between our (FAMILY) high and low was 66.4g, similarly the contract their high and low meat in the prudent group was 22.95g and ours (START) was 20.42g (see table 1 of response letter). 

In the Zhang et al. (2006) paper the magnitude of the RR was 1.63 (western) and 1.39 (prudent) in ours our high vs low was OR was 1.63 (FAMILY) and 1.14 (START). 

However, at the request of a previous reviewer we change the reference group from low to medium for ease of interpretation. 

In our study with 91/581 cases – FAMILY and 241/976 cases START, we have 80% power to detect an OR of at least 1.74 in FAMILY and, in START an odds ratio as small as 1.41. With 758/13,110 (5.8%) cases, Zhang et al. report a 95% confidence interval of the observed relative risk, comparing a western and prudent diet, to range from 1.09 to 2.21. We are sufficiently powered to detect associations consistent within this range.

We do not believe that a statistical test of interaction would be of value because we are not testing the hypothesis of between-cohorts interaction. We are assuming that the cohorts will be different because of their sample characteristics, and the distribution of the outcomes and exposure. This is why we decided to treat the cohorts separately, and not test for interaction formally.

However, the issue of sample size and power calculation is address in the discussion – limitation section of the paper:

“Third, our sample size was small, which reduced our power in multivariable analyses. Nevertheless, our point estimates in FAMILY are aligned with larger studies which have reported positive associations between red and processed meats and GDM [21,22,24]. We also had a relatively high event proportion or GDM (16% - FAMILY and 25% - START) compared to other studies which had approximately 5% [21,22,24,25].”

Page 26, lines 402 - 406

Comment 6

Lines 157 to 160 There isn’t sufficient explanation to say why the authors decided to analyse only data of white European from FAMILY cohort study especially there were south Asians in that cohort who were a major ethnic group the analysis is focusing on. 

Response 6

Thank you for this comment, we have included the following:

“The START cohort included South Asian Canadians, the largest non-white ethnic group in Canada while the FAMILY cohort included White Europeans, the largest ethnic group in Canada., although the FAMILY cohort was multi-ethnic [White European, East or South East Asian, Aboriginal, South Asian, and African or other origins] [28], the majority of participants (>80%) are White European, as a result we opted to exclude other ethnic groups (which includes approximately 2% South Asians) for our analysis” Page 7, lines 155 – 161.

Furthermore, only 2.1% (n = 17) in FAMILY was South Asian and 0% (n = 0) of White European in START. Within the FAMILY cohort we had only 5 cases of GDM of the 17 South Asian participants (before applying our inclusion and exclusion criteria). Given this we opted not to included South Asians of the FAMILY cohort in our South Asian population for this analysis. 

Sensitivity analysis of the difference in odds ratio if we were to include South Asian (15 once the study inclusion and exclusion criteria is applied) from the FAMILY in the START population revealed the results would have been similar (see tables 2a and 2b of response letter).

Comment 7

Line- 182-183 Recipes were generated for red and processed meat composite food items that were not available in the database, and a nutrient value/ profile was developed.

Response 7

N/A

Comment 8

Lines 233-236: Why did the authors decide to place 3 equally-size groups? They were basically comparing within the group. To be biologically plausible, wasn’t it good to place them in respective tertiles by defining the high, medium and low intake instead of cohort specific tertile cut-offs. 

Response 8

Thank you for this question. Food intake is not normally distributed (positively skewed; all values >=0) and by placing people into categories of red and processed meat intake we minimize the impact of outliers and ensure equal group size at each exposure level, and those optimize our power and stability of risk estimates within the groups.

Nutritional epidemiology has a long history with this approach and therefore our results will be more comparable to other studies.

The following has been added to the paper:

“By creating cohort-specific tertile cut-offs and acknowledging the large number of processed meat non-consumers in the START cohort we minimize the impact of outliers and ensure equal group size at each exposure level, and those optimize our power and stability of risk estimates within the groups.”

Page 11, lines 269 - 273 

Comment 9

Line 237-240: As more than 50% of the participants in START were non-consumers of processed meat, using cohort specific tertile cut-offs for the remaining 239 participants is not correct. Should have used a definition for high, medium and low intake to prove a biologically plausible association 

Response 9

Thank you for your comment within the manuscript it states:

Participants in each cohort were placed into three equally-size groups according to cohort-specific tertile cut-offs (refers to any of two points of the ordered distribution of consumers which divides the group into three equal parts) for consumption of unprocessed red meat and total red and processed meat (low, medium, and high). Processed meat was handled differently. In START, most participants (56.9%, n=565/976) reported consuming 0 g/d of processed meat. Therefore, we created a non-consumer group and placed the remaining participants into groups based on tertile cut-off points (low, medium, and high consumers). Pages 10, lines 263 – 269

The following has been added to the paper:

“By creating cohort-specific tertile cut-offs and acknowledging the large number of processed meat non-consumers in the START cohort we minimize the impact of outliers and ensure equal group size at each exposure level, and those optimize our power and stability of risk estimates within the groups.” Page 11, lines 269 - 273

Comment 10

In summary there are several methodological issues in this study. My recommendation is to reanalyse and revise the manuscript. 

Response 10

N/A

Editorial Comment 1 

Response to Editorial Comment 1

The following references have been added to the manuscript:

10. Berkowitz GS, Lapinski RH, Wein R, Lee D. Race/ethnicity and other risk factors for gestational diabetes. Am J Epidemiol. 1992;135: 965–973. 

11. Feig DS, Berger H, Donovan L, Godbout A, Kader T, Keely E, et al. Diabetes and Pregnancy. Can J Diabetes. 2018;42: S255–S282. doi:10.1016/j.jcjd.2017.10.038

12. Giannakou K, Evangelou E, Yiallouros P, Christophi CA, Middleton N, Papatheodorou E, et al. Risk factors for gestational diabetes: An umbrella review of meta-analyses of observational studies. PLOS ONE. 2019;14: e0215372. doi:10.1371/journal.pone.0215372

13. Mijatovic-Vukas J, Capling L, Cheng S, Stamatakis E, Louie J, Cheung NW, et al. Associations of Diet and Physical Activity with Risk for Gestational Diabetes Mellitus: A Systematic Review and Meta-Analysis. Nutrients. 2018;10. doi:10.3390/nu10060698

14. Schoenaker DAJM, Mishra GD, Callaway LK, Soedamah-Muthu SS. The Role of Energy, Nutrients, Foods, and Dietary Patterns in the Development of Gestational Diabetes Mellitus: A Systematic Review of Observational Studies. Diabetes Care. 2016;39: 16. doi:10.2337/dc15-0540

15. Teh WT, Teede HJ, Paul E, Harrison CL, Wallace EM, Allan C. Risk factors for gestational d

---

## [Decision Letter · Decision Letter 2]

1 Apr 2024

The association of red and processed meat with gestational diabetes mellitus: results from 2 Canadian birth cohort studies

PONE-D-22-22041R2

Dear Dr. de Souza,

We’re pleased to inform you that your manuscript has been judged scientifically suitable for publication and will be formally accepted for publication once it meets all outstanding technical requirements.

Kind regards,

George Kuryan

Academic Editor

PLOS ONE

Additional Editor Comments (optional):

Reviewers' comments:

Reviewer's Responses to Questions

**Comments to the Author**

1. If the authors have adequately addressed your comments raised in a previous round of review and you feel that this manuscript is now acceptable for publication, you may indicate that here to bypass the “Comments to the Author” section, enter your conflict of interest statement in the “Confidential to Editor” section, and submit your "Accept" recommendation.

Reviewer #2: All comments have been addressed

2. Is the manuscript technically sound, and do the data support the conclusions?

Reviewer #2: Yes

3. Has the statistical analysis been performed appropriately and rigorously? 

Reviewer #2: Yes

4. Have the authors made all data underlying the findings in their manuscript fully available?

Reviewer #2: Yes

5. Is the manuscript presented in an intelligible fashion and written in standard English?

Reviewer #2: Yes

6. Review Comments to the Author

Reviewer #2: The authors have revised the introduction with better clarity of the cohorts chosen, the composition of the cohorts and reason for choosing the respective cohorts. The authors have clarified all the queries raised in the methods and statistical analysis section. They have described the best possible methods used for dietary assessment and analysis of data using traditional methods in nutrition epidemiology with its own limitations. Though this study has negative results, it will make significant contribution to tease out the inconsistent association found between red meat, processed meat, unprocessed meat and GDM.

It can be considered for publication.

7. PLOS authors have the option to publish the peer review history of their article (what does this mean?). If published, this will include your full peer review and any attached files.

Reviewer #2: **Yes: **Rita Isaac

---

## [Editor Report · Acceptance letter]

9 May 2024

PONE-D-22-22041R2 

PLOS ONE

Dear Dr. de Souza, 

I'm pleased to inform you that your manuscript has been deemed suitable for publication in PLOS ONE. Congratulations! Your manuscript is now being handed over to our production team.

Kind regards, 

on behalf of

Professor George Kuryan 

Academic Editor

PLOS ONE